# A qualitative study to investigate pharmacovigilance systems in Dubai hospitals

Sawsan Shanableh[1], Muaed Jamal Alomar[1]*, Hadzliana Zainail[2]

1 Department of Clinical Sciences, College of Pharmacy and Health Sciences, Ajman University, Ajman, United Arab Emirates, 2 Discipline of Clinical Pharmacy, School of Pharmaceutical Sciences, University Sains Malaysia, Penang, Malaysia

* muayyad74@yahoo.com

## Abstract

### Background

Ensuring the safety of medications is a significant public health priority, with developed countries implementing robust pharmacovigilance programs. Despite this, healthcare providers continue to underreport adverse drug reactions (ADRs). This study aims to explore the existing pharmacovigilance system and procedure followed for ADR reporting in selected Dubai hospitals. It also identifies the obstacles that may negatively influence ADR reporting.

### Methods

This study was conducted using a qualitative methodology involving in-depth interviews. Convenience sampling was used to select and interview chief hospital pharmacists in Dubai using a semi-structured interview guide. Every interview was audio recorded, verbatim transcribed, and then subjected to a thematic content analysis. The data were analyzed manually by the reading and re-reading of the interviews, and an inductive and flexible approach was undertaken by the research team.

### Results

A total of ten individuals, three chief pharmacists and seven pharmacy managers were interviewed. Seven themes emerged from the interviews' thematic content analysis, including: (1) Existence of a pharmacovigilance center or unit, (2) Experience with medication safety & ADR concept, (3) Current ADR reporting process in the hospital setting, (4) Barriers to adverse drug reaction reporting, (5) Policy change needs, (6) The factors that could enhance ADR reporting, and (7) Future pharmacovigilance research needs in the UAE. Eight hospitals have ADR reporting systems in their hospital policies while two hospitals don't have any pharmacovigilance system. Of the eight hospitals, seven have their own software for ADR reporting and one used paper

**Data availability statement:** All relevant data are within the paper and its Supporting Information files.

**Funding:** The author(s) received no specific funding for this work.

**Competing interests:** No authors have competing interests.

form. Based on the participants feedback, six have full experience with ADR reporting process, while others have partial or don't have any idea about reporting process. ADR reporting is hindered by a number of factors, such as inadequate awareness and training for pharmacovigilance and ADR reporting process, a communication gap between hospitals and regulatory bodies, time constraints due to numerous responsibilities, and fear of punishment.

## Conclusion

The study found that although hospital pharmacists have a good understanding of medication safety and ADR reporting, they do not actually apply this knowledge into practice. Ongoing training and education on the ADR reporting process have the potential to improve attitudes toward ADR reporting and actively engage individuals in ensuring medication safety within hospital settings. The participation of key stakeholders from the Ministry of Health, academia, the pharmaceutical companies, and healthcare professionals is essential to support the safe and effective use of medications.

## Introduction

Adverse Drug Reaction (ADR) reporting is a key pharmacovigilance (PV) method aimed at ensuring medication safety and mitigating harmful events [1]. It is critical in detecting serious and rare ADRs early, guiding causality hypotheses, investigation priorities, and regulatory measures [2,3]. However, effective ADR reporting depends on the knowledge, attitude, and practices of healthcare providers [1,4–6].

Pharmacists, who are members of healthcare professionals, can play a significant role in ensuring the safety of medications because they have a thorough understanding of how drugs work and behave in the body, which enables them to identify potential ADRs [7]. The role of pharmacists has evolved significantly over time. Initially, their responsibilities were focused on the chemical composition and raw materials of drugs, drug production, and dispensing. However, in recent years, pharmacists have taken on a more consultative role, providing physicians with advice on drug therapy [8,9]. They play a critical role in reducing the incidence of ADRs and drug-drug interactions, as well as offering information and training on proper drug use [9,10]. In addition, pharmacists make significant contributions to pharmacovigilance as they are often the first to report adverse medication reactions. They are instrumental in the development, maintenance, and ongoing evaluation of ADR programs and are responsible for educating healthcare professionals about potential ADRs. Furthermore, pharmacists play a vital role in conducting post-surveillance safety and efficacy studies [11].

In the UAE, the role of hospital pharmacists is predominantly centered around medication supply and dispensing, with limited involvement in comprehensive patient care services. While they have the potential to extend their services to various

hospital areas, such as ambulatory care clinics, their activities primarily revolve around medication procurement, inventory management, and dispensing. Clinical assessment of patient needs for pharmaceutical care receives minimal attention, if any, due to factors such as insufficient clinical training and a shortage of pharmacists in hospital settings [12]. According to a study by Dameh conducted in 2009, these factors contribute to the challenges faced by hospital pharmacists in providing effective and efficient services [12].

The UAE's primary pharmacovigilance system, which is connected to the Uppsala Monitoring Centre and the WHO, comprises one national pharmacovigilance center that oversees the entire country [13]. Through the websites of the Ministry of Health (MOH) and other healthcare regulatory bodies in the UAE, the ADR reporting form is electronically accessible [13]. According to Beninger et al., all healthcare facilities in the UAE weather they are public or private were subject to the policy of reporting ADR [14]. Moreover, the members of the associated expert committees are in charge of creating circulars and publications on the ADRs, as well as analyzing and evaluating the factors causing ADR after receiving related reports [15]. In addition, Wilbur et al. have described the mandatory reporting of potential ADRs by UAE hospitals to national centers as well as the use of an assessment tool to calculate the likelihood that the medication caused the reported reaction [16]. Alomar et al. study, which sought to ascertain the knowledge, attitude, and practice (KAP) of ADR reporting within healthcare professionals in the UAE, discovered that KAP was lacking among these practitioners [17]. According to the findings, 81%, 83%, and 83.3% of physicians, community pharmacists, and hospital pharmacists, respectively, were uninformed that the UAE had an ADR reporting center, whereas 56%, 60%, and 72% were unaware of the proper reporting protocol [11]. Moreover, it was discovered that only 19%, 14%, and 12.1% of physicians, local pharmacists, and hospital pharmacists, respectively, reported adverse drug reactions (ADRs) [11]. Another research by Sathvik et al. revealed that 86.7% of doctors and 72.2% of pharmacists in the UAE were unaware of the ADR reporting system [18].

In this context, this study was conducted using qualitative methods to examine the knowledge, attitudes, and practices of hospital pharmacists in Dubai regarding Adverse Drug Reaction reporting. The qualitative approach fills gaps in the literature by using non-numerical data to investigate and comprehend the viewpoints and behaviors of the participants [19]. Despite the growing body of literature on pharmacovigilance, much of the existing research has relied on quantitative or survey-based methods, which primarily focus on measuring awareness levels, reporting rates, and compliance indicators. While such studies provide important statistical insights, they often fail to capture the underlying contextual, organizational, and behavioral factors that influence pharmacovigilance practices in real-world healthcare settings. In the context of Dubai hospitals, where healthcare systems are shaped by diverse institutional structures, multicultural workforces, and evolving regulatory environments, these dimensions are particularly significant. A qualitative approach is therefore essential to explore the lived experiences, perceptions, and systemic challenges faced by healthcare professionals involved in pharmacovigilance. This methodology enables a deeper understanding of the factors that support or hinder effective implementation, offering nuanced insights that cannot be obtained through standardized surveys alone. The findings from this research will contribute to the development of interventions to enhance medication safety.

To the best of our knowledge, this is the first qualitative study to assess hospital pharmacists' practices, knowledge, and potential obstacles to reporting adverse medication reactions in Dubai, UAE.

## Materials and methods

### Ethical approval

The study was approved by Dubai Scientific Research Ethics Committee (Approval Reference number: DSREC-SR-03/2022_02) and Universiti Sains Malaysia Research Ethics Committee (Approval Reference number: USM/JEPeM/22070495). Before the commencement of the study, participants completed an online consent form and were provided with a detailed briefing on the study's objectives. All interviews were audio-recorded with participants' consent, and the recordings were stored securely on a password-protected device accessible only to the research team. During the transcription process, all identifying information—such as names, job titles, and hospital affiliations—was anonymized

or removed to ensure confidentiality. Each transcript was assigned a unique code in place of personal identifiers. All data were managed in strict accordance with the guidelines approved by the institutional ethics review board, and every effort was made to maintain the highest standards of privacy and confidentiality throughout the research process.

## Study design

For this study, a qualitative research approach was chosen because it allows for an in-depth exploration of gaps that might be missed in survey-based research [20,21]. This approach will also provide a deeper comprehension of the participants' behavior and a thorough analysis of their thoughts and feelings to comprehend the circumstances surrounding ADRs reporting practice at the hospital levels [21,22]. It concentrates on non-numerical data, giving the researcher a comprehensive understanding of the respondent's perspective [22].

## Study setting

The research was conducted in Dubai, a city with a population exceeding 3 million [23]. Dubai serves as a prominent center in the UAE for culture, commerce, healthcare facilities, and medical tourism [24]. The study was conducted among pharmacy managers and head pharmacists who worked in different public and private hospitals in Dubai.

Dubai has a total of 28 hospitals, including 6 public and 22 private institutions [25]. Invitations to participate in the study were sent to all hospitals. Fifteen hospitals expressed willingness to participate. A purposive sampling strategy was employed to ensure a comprehensive representation of the pharmacovigilance landscape across both public and private sectors. The selected public hospitals were major tertiary care centers affiliated with the Dubai Health Authority (DHA), known for their large capacity and critical care services. Private hospitals were selected to reflect a diverse mix of ownership models (local, international, and free-zone), accreditation statuses (e.g., Joint Commission International [JCI] accreditation), and service capacities. Selection criteria included hospital type, bed capacity, level of pharmacovigilance activity, and willingness to participate. This approach enabled a balanced and contextually rich exploration of the factors influencing pharmacovigilance practices in Dubai. Ultimately, the study included two public and eight private hospitals.

## The creation of an interview guide

Based on a review of the literature, earlier researches, and typical pharmacovigilance practice in UAE [26–34], a semi-structured interview guide was created. The guide was created with the intention of exploring a variety of topics, including participant demographics, knowledge of the existence of pharmacovigilance centers in hospitals, perceptions and experiences with ADR reporting, general opinions about the UAE's ADR reporting system, obstacles to ADR reporting, suggestions for improving the country's current pharmacovigilance system, and future research needs in the area. The topic guides for the semi-structured interview are summarized in Table 1.

The validity and reliability of the guide were examined before any data were gathered. At Ajman University in Ajman, UAE, two expert professors and researchers validated the questionnaire. The study's reliability was guaranteed through in-person interviews with the participants. Conducting in-person interviews, as in this study, allows researchers to gather rich, detailed, and nuanced data directly from participants, minimizing the risk of misinterpretation and enhancing reliability. Furthermore, using a semi-structured interview guide ensures consistency by asking all participants similar questions, which supports comparability and reduces variability in data collection.

Two hospital pharmacy managers from two different hospitals in Dubai participated in a trial of the guide (who were excluded from the actual study). The researcher conducted a manual line-by-line analysis of the transcripts from the pilot study to identify pertinent themes and content. The pilot results were used to modify the guide accordingly. Following the experts' and interviewees' consent and comments, the study interview guide was established.

**Table 1. An overview of the semi-structured interview topic guide.**

| No | Topic Guide | Questions |
|---|---|---|
| 1 | Demographic Data | 1. Can you please introduce yourself? |
| | | 2. What is your qualification? |
| | | 3. What is your rank? |
| | | 4. Years of experience as a pharmacist? |
| | | 5. Country of graduation? |
| 2 | Existence of a pharmacovigilance center | 1. Is there a Pharmacovigilance Center or any other body assigned with the responsibility for monitoring ADR reporting in your working place? |
| | | 2. Does the Pharmacovigilance Center physically exist? |
| | | 3. Is there a clear mandate, organizational structure, roles, responsibilities, and reporting lines for the Pharmacovigilance Center? |
| | | 4. Is there an annual budgetary allocation for pharmacovigilance activities? |
| 3 | Experience with ADRs Reporting | 1. What is your experience with ADR reporting? |
| | | 2. What are the most common type of ADRs you used to see from your experience? |
| | | 3. All types of ADRs used to be reported? |
| 4 | General views on the ADR reporting system of the UAE | 1. Could you provide a detailed explanation of your hospital's ADR reporting procedures? |
| | | 2. Could you explain the workflow of ADRs reporting to national level? Do u have your own system (or system given by ministry)? |
| 5 | Barriers that affect ADRs reporting | 1. What obstacles might pharmacists face when it comes to practice PV and reporting ADRs? |
| | | 2. Can you share examples of cultural or organizational barriers that may prevent the effective reporting of adverse drug reactions? |
| | | 3. How do you think technology and data management systems contribute to or alleviate barriers in ADR reporting? |
| | | 4. From your standpoint, what changes in policy or regulation could help address and remove barriers to ADR reporting? |
| 6 | Views to improve the current pharmacovigilance system in the UAE | 1. How can we encourage hospital pharmacists to report ADRs? |
| | | 2. How can healthcare providers and institutions collaborate to create a more open and supportive environment for reporting ADRs? |
| | | 3. In your opinion, what role do patient perspectives and engagement play in overcoming barriers to ADR reporting? |
| | | 4. Are there specific educational or awareness initiatives that you believe could positively impact ADR reporting rates? |
| | | 5. How might a multi-stakeholder approach involving healthcare professionals, regulatory bodies, and pharmaceutical companies improve the overall ADR reporting process? |
| 7 | Future pharmacovigilance research needs in the UAE | 1. What kind of studies are we going to need in the future to improve the reporting of ADRs by UAE healthcare providers? |
| | | 2. To which extend do you think conducting continuous education sessions and training to pharmacists about ADRs reporting can enhance the process? |

**Study sampling and recruitment.** There is no set sample size for qualitative research; data should be collected continuously until data saturation is achieved [35]. As a result, the quantity of participants is determined by the quality of the data. According to Miles and Huberman's book on qualitative data analysis, having more than 15 cases in the sample size can make analysis challenging and complex [36].

Purposive and convenient sampling were used in conjunction with the researchers' contacts to recruit chief pharmacists in this study [37]. Taking into consideration their professional background, the targeted participants were chosen from Dubai's public and private hospitals. Participation in this study was voluntary for all chief pharmacists and pharmacy managers.

*(Continued)*

**Data collection.** The interview process took place between January and February of 2023. Before taking part in an interview, each chief pharmacist or pharmacy manager signed a written consent form. Each participant was made aware that the interview will be recorded and used for publication in future. The main researcher who has received training in qualitative interview techniques, conducted all of the interviews, which lasted between 20 and 45 minutes on average. The interviews took place at the participants' place of employment as per their convenience. English was the language used for the interviews. Every interviewee receives the same set of open-ended questions and when more information was needed, appropriate follow-up questions were posed. After every interview session, they were also allowed to share any further opinions they had on the subjects covered.

After eight interviews, no new themes emerged, but two more interviews were carried out to confirm that saturation had been reached of a total number of ten interviews.

## Data analysis

Each interview was recorded, transcribed word by word using a Rev application [38], analyzed thematically using a method previously outlined by Braun and Clarke [39]. The initial step involved familiarizing oneself with the gathered data. After that, the transcripts were organized and arranged in a systematic way, which enabled the researcher to identify and codes the themes. Only the relevant themes or codes were extracted from the transcripts. By systematically reviewing the transcripts, relevant passages were highlighted and matched with appropriate codes. No predefined codes were employed; instead, each code was developed and refined during the data analysis process. A consensus was reached by all members of the assigned research team, and the co-authors were assigned to read the transcripts of the interviews to ensure that the themes and codes generated accurately reflected the content of the interviews. Table 2 presents a summary of the various stages of analysis.

## Reporting

The utilization of the COREQ (Consolidated Criteria for Reporting Qualitative Research) checklist facilitated the comprehensive reporting of both methods and results in this qualitative study [40].

## Inclusivity in global research

Additional information regarding the ethical, cultural, and scientific considerations specific to inclusivity in global research is included in the Supporting Information (Inclusivity in global research).

## Results

### Demographic data of participants

A total of ten individuals, three chief pharmacists and seven pharmacy managers were interviewed. There were two females and eight males present. They were between the ages of 40 and 49. Of the participants, five held a Master's or PhD in their field of specialization (Pharmacy), and the remaining five were graduates. Three pharmacists had 10–14

**Table 2. Summary of data analysis process for the qualitative phase.**

| Stage of Analysis | Tasks Fulfilled |
|---|---|
| Stage 1: Familiarization with data | Transcribing interview transcripts, reading, and rereading. |
| Stage 2: Initial creation of codes | Set up codes for all data |
| Stage 3: Look for themes. | Sorting codes into possible themes |
| Stage 4: Overview of the themes | Assuring the themes' external heterogeneity and internal homogeneity. |
| Stage 5: Identifying and classifying themes | Additional theme improvement |
| Stage 6: Report completion | Write up and choose representative quotations |

years of experience, while seven pharmacists had 15 or more years of service experience. The demographic details of participants are outlined in Table 3.

**Thematic analysis**

The thematic content analysis of the interviews resulted in seven major themes. The main themes were:

1. Existence of a pharmacovigilance center or unit

2. Experience with medication safety & ADR concept and reporting

3. Current ADR reporting process in the hospital setting

4. Barriers to adverse drug reaction reporting

5. Policy change needs

6. The factors that could enhance ADR reporting

7. Pharmacovigilance research needs in the UAE

**Table 3. Socio-demographic characteristics of the pharmacists (n = 10) – Phase 2.**

| Demographic Characteristics | Numbers |
|---|---|
| **Gender** | |
| Female | 2 |
| Male | 8 |
| **Age** | |
| 30–39 | 2 |
| 40–49 | **7** |
| ≥ 50 | 1 |
| **Qualification** | |
| Bachelor in Pharmacy | 1 |
| PharmD | 4 |
| Master in Pharmacy | 4 |
| PhD | 1 |
| **Country of Graduation** | |
| UAE | 1 |
| Other Countries: | |
| - Middle East (Saudi Arabia, Kuwait, Oman, Jordan, Syria, Lebanon, Iraq, Iran) | 2 |
| - Asia (India, Pakistan, Philippines, Bangladesh) | 3 |
| - Africa (Libya, Algeria, Egypt, Sudan, Somalia) | 1 |
| - America & North America (USA, Canada) | |
| - Europe (UK, Cyprus) | 3 |
| **Current Position** | |
| Chief Pharmacist | 4 |
| Pharmacy Manager | 6 |
| **Years of Experiences** | |
| 10–14 | 3 |
| ≥ 15 | 7 |

**(a) Theme 1: Existence of a pharmacovigilance center or system**

 **(a)(i) Subtheme 1: Presence and Familiarity of PV center or system in hospital.** During the interviews, participants were asked about the presence of pharmacovigilance centers or systems within their workplaces. Eight hospitals have ADR reporting systems in their hospital policies while two hospitals don't have any pharmacovigilance system. Of the eight hospitals, seven have their own software (e.g., OVR, RL, PV Portal) for ADR reporting and one used paper form.

*"Yeah, in my current hospital, still now we are manually filling the ADRs happened using ADR-reporting form"* **(P 1)**

*"Well there is a reporting system in our hospital we have called the OVR and it's merged with the medication error portal. But not specific system for the ADR alone"* **(P 2)**

*"There is no dedicated center with the name of pharmacovigilance in our hospital. As I told you that our quality and patient safety department, they are all dealing with such kind of things"* **(P 4)**

*"Here in AL Jalila we have RL system, which is events reporting system"* **(P 6)**

*"….So they report through the MER portal or the OVR portal and it by default goes to the quality"* **(P 9)**

Among the ten participants interviewed, six pharmacists were familiar with the reporting process flow, while the remaining four pharmacists lacked a clear understanding of the ADR reporting process in their respective workplaces.

*"Yeah, in my current hospital, once the patient complains either himself proactively or through investigation with the physician round or even to the nurse that one who witnessed that ADR should fill form ADR form and send it to the head of pharmacy where the head of pharmacy and his team will review that one and see its evidence and then go to the next level which is reporting to the authority if needed and also to the BTC for they are meeting every three months or monthly and if needed also. So to take action if needed action for example if this is really serious and we need to withdraw this drug or it's just a side effect commonly happen"* **(P 1)**

*"So when the nurse is the person who is administering the medication for example, so if she finds any complaint from the patient's side, for example, patient complains after administering some sort of itching or irritation. So the first thing she does stop the medication, report to the doctors and stabilize the patient if it is required. And once the patient is stable, so then they have to report it through the OVR portal. Once she completes the OVR portal, it goes to quality and the risk management department, which is called QRM, there is one person risk manager assigned for all sort of errors"* **(P 2)**

*"….So once you administer the medication at the inpatient level or at the hospital level. So we need to monitor and nurses will monitor the effects of the medication"* **(P 10)**

 **(a)(ii) Subtheme 2: Presence of designated person to report ADR.** The hospital pharmacists were also questioned about the presence of a designated individual for ADR reporting duties. Within the eight hospitals possessing an ADR reporting system, the pharmacy managers are incharge of the hospital's PV activities. There are no certified or professional pharmacists incharge of PV and ADR operations.

*"No, we don't have one specialized, we don't have center even or one person who's specialized for this task"* **(P 1)**

*"Correct. I am the one responsible for ADR reporting at the hospital as I am the Clinical Pharmacy manager"* **(P 2)**

*"I am the clinical pharmacist in the hospital, I am the one responsible to analyze the ADRs, medication errors, near misses in coordination with the equality risk management department"* **(P 5)**

*"Yes. Previously in my previous hospital, we have medication safety officer who is responsible for awareness and evaluation of any events. Now in this hospital, I'm doing this service in addition to other clinical pharmacists"* **(P 6)**

*"To check the ADR activities, of course the head of the pharmacy is the person in charge"* **(P 10)**

**(a)(iii) Subtheme 3: Presence of PV activity budget.** Additionally, it was also found that no budget was allocated for PV and ADR reporting activities in any hospitals. It is impeded within the pharmacy budget.

*"No, there is no any annual budget allocated for PV activities. If there is any quality or safety awareness week in the hospital, then we do some activities related to PV in that week. But not putting any budget specifically for pharmacovigilance, no"* **(P 2)**

*"No. No, there is no separate budget for ADR reporting activities"* **(P 3)**

*"There is no specific budget for pharmacovigilance activities at the hospital"* **(P 7)**

**(b) Theme 2: Experience with medication safety & ADR concept and reporting**

**(b)(i) Subtheme 1: Experience with ADR reporting process.** Based on the participants feedback, six have full experience with ADR reporting process, three have partial experience, especially with the differentiation between ADR and medication errors and one was not aware of the reporting process.

*"I have quite experience in all this hospital I mentioned. So in Dubai Hospital for example when we were in Dubai Hospital there was a system, it's called Aman System and where we used to report any incident and one of them is the ADR report and it's online and it'll go to all head of concern department and if they have any comment they will add that in the loop and everybody will be aware about that. "* (P 1) (Full experience)

*"And my experience seeing this is very under-reporting, the numbers are coming one, two or three per month, which is not the real case actually there are many adverse drug events related or adverse drug reaction which goes unnoticed and not reported"* (P 2) (Partial experience)

*"In UA E, what is happening for example, if you will talk in within the pharmacy? Mostly we are tracking drug adverse reaction by pressure of clinic. In the pharmacy we are tracking adverse drug reaction by clinical trace drugs."* (P4) (No experience)

**(b)(ii) Subtheme 2: Type of ADR to be reported:.** When asked about the preference (type and/or severity) of the ADR to be reported, the participants gave a variety of answers; only major and severe ADRs can be reported according to some pharmacists. While some stated that any ADR, no matter how mild, have to be recorded. Certain pharmacists stated that while recognized ADRs do not need to be reported, rare and uncommon ones should.

*"If it is in the leaflet of the drug, is it part of the common adverse effect and it's already well known so such ADR I don't think we need to report because it is well known side effect of such a drug. But the thing which is less likely to happen and that will be discovered during the drug available for large scale, especially for the new drug, if there is a new drug in the market which is not yet studied only on a small scale and now when we go to the large scale there is something which were rare, it can appear more with more patient. That is very important to report it"* **(P 1)**

*"Yes, the policy talks about this that we should report allergic reaction and other serious and major ADR, but the other type of the adverse drug events, it is not mandatory"* **(P 2)**

*"We encourage all healthcare providers at the hospital to report anything related to medications. Any type of ADRs and medication errors regardless if they are major or minor, rare or common. Then further investigation will come through the pharmacy"* **(P 6)**

*"Only major and severe ADRs can be reported. No need to report the common ones"* **(P 9)**

*"We do report all types of ADRs regardless if they are major or minor, severe or mild. Rare or common. All ADRs and medication errors we do report them"* **(P 10)**

Each hospital independently determined what information to report to the Ministry of Health, even though, there is a policy mandating that severe ADRs must be reported to the MOH within 48 hours.

*"So we don't believe there's an necessary to report everything. But if there is something like a new side effect which happened first time, definitely we'll report it to the MOH and we'll see what's the response. Definitely they have a system in place but we did not have this experience so far".* **(P 1)**

*"All types, minor, major, serious, all types of ADRs we reported to DHA on monthly basis"* **(P3)**

*"10 days up to my knowledge, the serious issue to be reported. Okay. Especially if it is a new drug within 24 hours….. Yes. Internally we did the reporting, everything should be reported just for a purpose of improvement, but not for the ministry of health"* **(P 6)**

*"I think it should be less than 48 hours, the maximum. But for DHA, I'm not sure about the rules"* **(P 7)**

*"If it needs to record, like serious one, then yes, of course we reported to MOH"* **(P 9)**

**(c) Theme 3: Current ADR reporting process in the hospital setting**

While the majority of hospitals have some sort of ADR reporting system, only six have implemented a comprehensive ADR reporting procedure.

**(c)(i) Subtheme 1: Pharmacists' accountability for hospital ADR reporting.** Based on the feedback from the four participants who had unclear ADR reporting process, pharmacists are not involved in the analysis and follow-up on ADR reports; this is a duty of pharmacy managers.

*"I am the one responsible for ADR reporting at the hospital as I am the Clinical Pharmacy manager"* **(P 2)**

*"I am the one responsible of ADR reporting system at the pharmacy as other pharmacists don't deal with this system directly and they don't know how to do reporting"* **(P 3)**

*"In the current hospital we have actually me as a pharmacy manager to be responsible. I am the one who reporting as an quality performance indicator of this one. As other pharmacist in the pharmacy have other tasks and they don't know how to fill the report of ADR"* **(P 9)**

**(c)(iii) Subtheme 2: The simultaneous presence of the MOH system and hospital ADR reporting system.** Based on the interviews and the feedback from the participants, the seamless transfer of ADR reporting between hospitals and the Ministry of Health is not evident. None of the reporting systems in the hospitals were directly connected to the MOH of the UAE or to the Dubai Healthcare Authority (DHA).

*"It's not very clear to be honest. I have seen one memo in the beginning initially. So we are under the Dubai Health Authority and there is one circular mentioning that we need to report the ADRs and there is the timeline"* **(P2)**

*"we need to report in addition to that quarterly reporting is being done by QRM to DHA on the medication error and the ADRs including All this"* **(P 5)**

*"Now we have two type of medications here in Dubai, medication which is registered in UAE; we either notify the supplier about any adverse drug reaction happen or the company itself, for medications registered in the ministry, we report it to the MOH if its serious within 48 hours"* **(P 6)**

*"It's not connected directly to DHA or MOH. Our QPS department, our quality and patient safety department is sending data to DHA about medication errors and serious adverse regulation happened"* **(P 10)**

**(d) Theme 4: Barriers to adverse drug reaction reporting**

There were numerous obstacles leading to ADR underreporting.

**(d)(i) Subtheme 1: Workload and manpower at work.** The majority of respondents claimed that a variety of reasons, such as a lack of time, limited number of staff, and unlimited tasks given to pharmacists are the main obstacles to prevent them from reporting ADRs at the hospital level.

*"…number one is limited number of staff and time. So time is always the big obstacle. So we have limited time and we have unlimited tasks and if we give the people task more than that, they cannot do it and it'll keep accumulation. This is obstacle number one. So there is no specific person assigned to do the ADR follow-up"* **(P 1)**

*"….But so this requires really big time to review the patient cases which most of the pharmacists in their busy routine will not find this time. So the time management here to review the patient cases to find out if there is any adverse drug reaction is a big obstacle"* **(P 2)**

*"The biggest challenge to be honest with you is number one is availability of the staff. If staff is less and work is more, then people will not care to report"* **(P 4)**

**(d)(ii) Subtheme 2: Lack of experience in reporting and lack of a formal reporting system.** Some of the participants also recognized the absence of an electronic reporting system, inadequate knowledge or training regarding the reporting process, types of ADRs, and the significance of PV activities, as well as the lack of a structured reporting mechanism, as barriers to ADR reporting.

*"Number two, maybe the lack of the training about ADR reporting. Maybe we need more training for staff who will work on this area to be more specialized and to differentiate between side effect and ADR, et cetera. And also to be formalized with the system"* **(P 1)**

*"Number two thing is that lack of awareness and training. For example, if one staff is coming from India or Pakistan and they're from the remote areas, they don't care about these things"* **(P 4)**

*"Absence of electronic system can be an obstacle to practice ADR reporting"* **(P 9)**

**(d)(iii) Subtheme 3: Lack of support from a management perspective.** Other factors cited by the participants as obstacle to ADR reporting were the lack of support from managerial level and colleagues, poor communication amongst medical professionals, and the fear of legal liability.

*"…the reporting of the medication errors or ADR on the OVR portal, sometimes we as a pharmacist feel that it's taken as a personal between the team as well and being worried from the consequences"* **(P 2)**

*"So the obstacles I would say in most of the cases is the poor communication amongst medical professionals. For example, if I talk about outpatient or inpatient, you are not dealing directly to the patient at the level of administration of medications. So if we have a clinical pharmacist they are having around so they can know directly if any sort of ADRs are happening, if it is missed from the nursing side. So the pharmacists cannot know"* **(P 3)**

**(e) Theme 5: Policy change needs**

The participants were asked if they thought that any adjustments to the hospital's policies or procedures could improve the practice of reporting ADRs.

**(e)(i)  Subtheme 1: Establishing PV center in each hospital.**  They all mentioned that setting up a PV center in every hospital and designating a specialist for this role would enhance ADR activities.

*"….so maybe we need office specified for the person who will be working for pharmacovigilance. He will be in his office with the resources to support him, different resources, references, internet, computer, access to the literature, all this. If we have and person specialized to do that, who will trained in this area, specialized in this area, definitely will be moving the pharmacovigilance from one level to second level and higher level"* **(P 1)**

*"Maybe first thing we need, is to establish PV center in each hospital with a pharmacist specialized for this job. Then train this person how to deal with ADR reporting system at the hospital level and maybe he will be the person assigned later on to train the other staff"* **(P 4)**

*"Another important thing is to force each hospital to establish PV center with staff dedicated for ADR reporting practice at the hospital level"* **(P 9)**

**(f) Theme 6: The factors that could enhance ADR reporting**

The hospital pharmacists were also invited to share their perspectives on the factors that might improve ADR reporting and PV system.

**(f)(i)  Subtheme 1: Continuous training on ADR reporting.**  Every participant recommended that pharmacists and/or other healthcare professionals be given educational training on ADR reporting, and that they periodically follow it up with additional information.

*"I believe training sessions; continuous education is very important here. Even if you have given one time then you need to keep them realizing the importance of reporting by continuous ongoing education. This will definitely improve the reporting. I have noticed myself, so one time the staff were oriented, in that month some good reporting happened and then later on when there are no follow up educations, you will see that the trend of reporting going down"* **(P2)**

*"Continue education and update of the knowledge related to types of ADR and medication errors, the process of reporting all can enhance the ADR practice"* **(P 3)**

*"I would say training sessions and continuous education for all healthcare providers on this topic will enhance the ADR reporting practice"* **(P 4)**

*"First providing continuous education about ADR reporting for all healthcare providers"* **(P 8)**

*"There should be a campaign or there should be an educational activity about PV to be given for all physicians, nurses and pharmacists"***(P 9)**

**(f)(ii)  Subtheme 2: Train the newcomers in the hospital for ADR reporting.**  Additionally, they suggested that all newcomers, regardless of their ranking, should receive this kind of training at all times.

*"When the new staffs are coming, the new manpower you have, the new human resources, so they need to be oriented also to make themselves aware what is the policy on the pharmacovigilance and the reporting etc especially for the people came from outside the country"* **(P 2)**

*"For any pharmacist, he's joining new here, first an orientation about medication safety measures in the hospital has to be given to him"* **(P 6)**

*"When anybody is joining the hospital, they should have training about PV system and ADR reporting"* **(P 7)**

**(f)(iii)  Subtheme 3: Provide a dedicated budget for the operations of the PV center.**  Establishing a PV center at every hospital and allocating a distinct budget for these kinds of initiatives were among other suggestions made by the participants.

*"…there should be a single person responsible for this program maybe with co-share, maybe physician plus pharmacist to be the core member and the co-leader for that program. Then other people like IT, authorities, nurse, PTC representative, all this is as a member to help in establishing this and then reporting"* **(P 1)**

*"Introduce a medication safety hotline or like a center for the vigilance or dedicated person. So for that you need to assign a dedicated person, you can name him anything medication safety officer or pharmacovigilance officer for example"* **(P 2)**

*"Assign budget for the PV activities"* **(P 8)**

*"Another important thing is to force each hospital to establish PV center with staff dedicated for ADR reporting practice at the hospital level"* **(P 9)**

*"…there should be maybe budget assigned for such work. Because really anything need work, it needs time and if it needs time, it needs people and if it needs people it need money at the end of the day"* **(P 10)**

**(f)(iv)  Subtheme 4: Legal protection for pharmacists by their workplace.**  A few pharmacists also brought up the subject of legal responsibility that they could be threatened at any time. Hence motivation and feeling secure would encourage reporting.

*"So giving them self-realization that their reporting can improve the process of medication safety, the patient outcomes and if they have reported this will be taken seriously and the process will be improved, not the persons to be made responsible for these things"* **(P 2)**

*"Secondly, I would say that we can encourage the reporting only when we make sure that this reporting is non-punishment. Okay. The staff and staff, they should be encouraged enough and they should be considering this reporting is a process to improve the medication safety, improve the gap and they should not feel that they will be punished for this"* **(P 5)**

*"We have different ways of encouraging ADRs reporting, one way we follow the non-blame system. Non-blame system means that our concern is the system, not the one who make the issue"* **(PH 6)**

**(f)(v)  Subtheme 5: Rewards for pharmacists do reporting.**  Another recommendation made by the participants was to give incentives, certificates of appreciation, and rewards to those who provide the most ADR reports.

*"Maybe we need to make an incentive for the people who report more so the top reporter for ADR in the organization. Maybe we can give him certificate in quality day that he's the highest number of reporting ADR or who prevent more ADR of that etc."* **(P 1)**

*"…putting an incentive will encourage the pharmacist to report it"* **(P 2)**

*"Yes, proper training, reward and recognition system and make them aware about the significance of reporting ADRs"* **(P4)**

*"Then, give a rewards or incentives to the one who is reporting the most every month"* **(P 10)**

**(f)(vi)  Subtheme 6: Engage patients in ADR reporting.**  Certain pharmacists recommended to involve patients and encouraging them to report any ADR immediately whenever they occur.

*"And the reporting can happen at the level of prescribing itself, dispensing and administration and even the patient themselves"* **(P 2)**

*"And finally, encouraging the patients themselves to report can enhance the ADR reporting practices.* **(P 8)**

*"Everyone has to be responsible for reporting. Nurses, pharmacists, physicians and even patients"* **(P 9)**

*"So when the patients and the family for example also be involved in ADR reporting, this can enhance the PV activities"* **(P 10)**

**(g) Theme 7: Pharmacovigilance research needs in the UAE**

The participants were also asked to give their views on the future research needed in the UAE that could serve the PV activities in the country and enhance ADR reporting.

**(g)(i)  Subtheme 1: Pharmacoeconomic studies.**  In their opinion, further pharmacoeconomic studies are necessary to determine the pharmacoeconomic burden of underreporting.

*"Do pharmacoeconomic study to calculate the amount of money missed or lost because of unreported idea. This will really make people think how much we are losing because of ADR"* **(P1)**

*"For that, maybe we need a study to calculate the cost we lose in case we don't do reporting"* **(P 4)**

*"Calculate the money we lost in case we are not reporting"* **(P 7)**

**(g)(ii)  Subtheme 2: Most prevalent forms of ME and ADR occurred.**  Others recommended doing research to identify the most common types of medication errors and ADRs happened and the ones which were not reported.

*"I believe the trend and types of most ADRs happened should be reviewed"* **(P 2)**

*"What are the difference between ADRs and medication errors"* **(P 8)**

*"What is the difference between medication errors and ADRs?"* **(P 9)**

*"Maybe we need to find the real number of missed ADR. We need to do a study to look how many ADR and medication errors were not reported"* **(P 10)**

**(g)(iii)  Subtheme 3: Using technology to enhance ADR reporting.**  Some participants recommended doing research on how technology may enhance ADR reporting. Additionally, they suggested to conduct studies to identify the real obstacles that hindered ADR reporting practice in the country.

*"We should find smart ways of reporting. So we need studies that highlighted this point and give suggestions of using technology to improve reporting"* **(P 2)**

*"The main thing is to find the reasons for underreporting in the country. How the technology could enhance PV activities in the country. Maybe to find the obstacles for underreporting in the country"* **(P 10)**

## Discussion

To the best of our knowledge, this is the first qualitative study that explores the current pharmacovigilance system and procedure for ADR reporting in Dubai hospitals. It also identifies the obstacles that may negatively influence ADR reporting and highlights the gaps that remain unnoticed by quantitative or survey-based methods [41]. This study contributes to a deeper understanding of pharmacovigilance by offering real-world insights into contextual challenges and behaviors, as recommended in qualitative health research [42]. Our findings revealed that not all pharmacy managers possessed a comprehensive understanding of the ADR reporting process, despite it being an integral part of their professional responsibilities. This knowledge gap is a significant barrier and is consistent with previous research indicating that poor understanding of ADR guidelines contributes to underreporting and increases the risk of medication errors [43–45]. According to the Theory of Planned Behavior, such knowledge deficiencies could negatively influence pharmacists' attitudes and perceived control over reporting practices, thereby reducing reporting rates [46]. This particular identified information must be considered when implementing interventions to enhance the reporting of ADRs.

Although more than half of the participants had encountered at least one ADR in their practice, they preferred to refer patients to physicians, indicating a lack of confidence in reporting independently. This mirrors findings from a study conducted in Lithuania, which highlighted that pharmacists could recognize ADRs but felt less confident in managing or reporting them [44]. Similar reservations are shared by nurses, reflecting broader interprofessional hesitancy in pharmacovigilance roles [47]. This attitude needs to be changed by implementing an educational intervention focused on the role of pharmacists in ADRs reporting.

Additionally, confusion between medication errors, side effects, and ADRs, as well as inadequate familiarity with the types of ADRs that require reporting, indicates a fundamental knowledge gap. Previous studies have confirmed that limited understanding of pharmacovigilance concepts and national reporting guidelines, such as those issued by the UAE Ministry of Health, significantly impairs the effectiveness of reporting systems [43,44,47]. The development of targeted training programs that integrate theoretical knowledge with practical application is imperative.

A commonly cited barrier in our interviews was pharmacists' high workload and limited time, which aligns with other studies reporting that time constraints and administrative burdens are among the main inhibitors of ADR reporting [44,47–49]. Organizational pressures and competing tasks often deprioritize reporting, which underlines the need for strategies such as streamlining documentation processes, utilizing electronic tools, and improving time management [47].

Lack of ongoing education and training was another significant barrier. Our participants emphasized the need for continuous professional development that includes up-to-date pharmacovigilance content. Previous literature supports this view, showing that sustained educational engagement correlates with increased ADR reporting [50,51]. Moreover, motivational factors—such as recognition, feedback, and supervisory encouragement—have been found to foster a culture of reporting [4,52–58]. This highlights the importance of a regular follow-up with pharmacists after any educational intervention to consistently remind them of reporting.

Most hospitals in the study had internal ADR reporting systems, typically supported by proprietary software. Prior research has highlighted that user-friendly digital platforms substantially enhance reporting practices [44,59]. Where access to such systems is limited, alternative methods—such as email templates, hyperlinks, or smartphone apps—can facilitate efficient ADR submissions [60]. However, the lack of awareness among pharmacists about the UAE's official smart reporting tool, UAE RADR, introduced by MOHAP in 2019 [61], suggests that promotional and training efforts around such tools remain insufficient.

Participants proposed establishing in-hospital pharmacovigilance centers and designating specific personnel for ADR reporting, supported by budget allocations from upper management. These recommendations are aligned with previous

studies suggesting that institutional support, dedicated drug safety units, and interprofessional collaboration are key enablers of effective reporting [44,62]. Enhancing pharmacists' perception of safety and institutional backing can significantly improve engagement in reporting.

Pharmacists also suggested incentive mechanisms such as certificates or awards for the most frequent reporters. Reward-based systems have been previously shown to positively impact ADR reporting behaviors [63]. Additionally, the potential role of patients in direct ADR reporting was raised. This aligns with European recommendations advocating for public involvement in countries with evolving pharmacovigilance infrastructures [48,64]. Given the low awareness of such mechanisms among Dubai's public, tailored awareness campaigns and public-facing digital tools could enhance reporting rates [13].

When asked about the future's researches, participant highlighted the topic of pharmacoeconomic burden of underreporting. This will improve the economic value of pharmacovigilance by including patients in the system. Further economic evaluation research is essential in this part of the world inorder to magnify the economic impact of detecting, preventing and reporting ADRs. A study conducted on 3012 patients have found that the detection of ADRs reduced the economic burden on healthcare and on the institution [65].

## Conclusion

In conclusion, many important issues have been underlined during this study. Lack of pharmacists' deep understanding of pharmacovigilance and its importance could impact the outcomes of any system even if it's available in the institution. It is recommended that structured and continuous education about this system is to be adopted as per institution separately and on the health authority level too inorder to build a constructive collaboration within and between healthcare institution under the umbrella of the Ministry of Health. It is also necessary to include the comprehensive process of ADR reporting, the clinical skills required for detection and the economic impact of preventing ADR while persuading the educational intervention. Further studies are necessary to pinpoint the actual and potential obstacles of implementing an ADR reporting system in the UAE and finding a way to overcome these problems. Once these problems are discovered they can be communicated to higher institutions and the Ministry where the impact of the interference will be fruitful and constructive.

### Limitations

Sampling technique could be one of the limitations in this study because it can be only generalized among that specific geographical area. The variation of the clinical experience among the participants with different clinical skills could slightly impact the results of the study and also the different ethnic backgrounds which carries with it a difference in languages could be a factor that affect their abilities to express their views in the best way possible.

## Supporting information

**S1 File. PLOS- Anonymized Participants.**
(ZIP)

**S2 File. Supportive data.**
(ZIP)

**S3 File. Inclusivity in global research.**
(DOCX)

## Author contributions

**Conceptualization:** Muaed Jamal Alomar, Sawsan Shanableh, Hadzliana Zainail.

**Formal analysis:** Sawsan Shanableh.

**Methodology:** Muaed Jamal Alomar, Sawsan Shanableh, Hadzliana Zainail.

**Project administration:** Sawsan Shanableh.

**Writing – original draft:** Sawsan Shanableh.

**Writing – review & editing:** Muaed Jamal Alomar, Hadzliana Zainail.

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
