## [Decision Letter · Decision Letter 0]

10 Jun 2025

PONE-D-25-10627A Qualitative Study to investigate pharmacovigilance systems in Dubai hospitalsPLOS ONE

Dear Dr. Alomar,

Thank you for submitting your manuscript to PLOS ONE. After careful consideration, we feel that it has merit but does not fully meet PLOS ONE’s publication criteria as it currently stands. Therefore, we invite you to submit a revised version of the manuscript that addresses the points raised during the review process.

We look forward to receiving your revised manuscript.

Kind regards,

Yaser Mohammed Al-Worafi

Academic Editor

PLOS ONE

Journal Requirements:

[NO authors have competing interests].

5. Please amend the manuscript submission data (via Edit Submission) to include authors Sawsan Shanableh and Muaed Alomar.

6. Please amend your authorship list in your manuscript file to include authors Sawsan Deeb Shanableh and Muaed Jamal Alomar.

7. Your ethics statement should only appear in the Methods section of your manuscript. If your ethics statement is written in any section besides the Methods, please delete it from any other section.

Reviewers' comments:

Reviewer's Responses to Questions

**Comments to the Author**

1. Is the manuscript technically sound, and do the data support the conclusions?

Reviewer #1: Yes

Reviewer #2: Yes

2. Has the statistical analysis been performed appropriately and rigorously? 

Reviewer #1: Yes

Reviewer #2: Yes

3. Have the authors made all data underlying the findings in their manuscript fully available?

Reviewer #1: Yes

Reviewer #2: Yes

4. Is the manuscript presented in an intelligible fashion and written in standard English?

Reviewer #1: Yes

Reviewer #2: Yes

5. Review Comments to the Author

Reviewer #1: Strengthen the rationale for a qualitative approach by highlighting gaps in previous quantitative or survey-based studies in the introduction part.

No details on the demographic distribution of public vs. private hospitals.

The discussion reiterates many points from the results section without sufficient theoretical grounding or linkage to wider literature.

Explicitly mention limitations inherent to the design and context, such as the exclusion of non-hospital settings or regulators.

Ethics approval is mentioned, but there is no mention of how confidentiality was maintained during audio recording and transcription.

Reviewer #2: 1. The study uses convenience and purposive sampling limited to only 10 participants, mostly male, and primarily senior positions (chief pharmacists and managers).

2. The study focuses solely on hospitals in Dubai, limiting applicability to other Emirates or healthcare sectors in the UAE.

3. Underdeveloped Thematic Framework: Although seven themes are outlined, some subthemes are repetitive or overlapping, and coding transparency is minimal.

4. Language and Grammar Issues: There are numerous grammatical errors, awkward phrasing, and redundancies throughout the manuscript. Conduct professional language editing to enhance readability and academic tone.

5. Minimal Integration of Patient and Public Role: Although patient reporting is briefly mentioned, the manuscript lacks substantive exploration of patient or caregiver involvement in pharmacovigilance.

6. Ethics and Consent Process Description: While ethical approvals are mentioned, the informed consent process and interview setting details (e.g., potential coercion, power dynamics) are not fully elaborated.

6. PLOS authors have the option to publish the peer review history of their article (what does this mean? ). If published, this will include your full peer review and any attached files.

**Do you want your identity to be public for this peer review?** For information about this choice, including consent withdrawal, please see our Privacy Policy .

Reviewer #1: No

Reviewer #2: **Yes: ** Omer Al-lela

---

## [Author Response · Author response to Decision Letter 1]

5 Aug 2025

Response to Reviewers

Response to Editor Comments

No Reviewers’ comments Changes

1 Please ensure that your manuscript meets PLOS ONE's style requirements, including those for file naming. The manuscript has been formatted to PLOS One style requirements.

2 Please include a complete copy of PLOS’s questionnaire on inclusivity in global research in your revised manuscript. Our policy for research in this area aims to improve transparency in the reporting of research performed outside of researchers’ own country or community. The policy applies to researchers who have travelled to a different country to conduct research, research with Indigenous populations or their lands, and research on cultural artefacts. The questionnaire can also be requested at the journal’s discretion for any other submissions, even if these conditions are not met. The manuscript includes a copy of the questions asked during the interview in the methodology section, under the title “The creation of an interview guide” (Table 1, page 8). This study was conducted in the author's country.

3 Please complete your Competing Interests on the online submission form to state any Competing Interests. It's been completed online.

4 Please confirm at this time whether or not your submission contains all raw data required to replicate the results of your study. Authors must share the “minimal data set” for their submission. I confirmed.

5 Please amend the manuscript submission data (via Edit Submission) to include authors Sawsan Shanableh and Muaed Alomar. It has been corrected as required

6 Please amend your authorship list in your manuscript file to include authors Sawsan Deeb Shanableh and Muaed Jamal Alomar. It has been corrected as required

7 Your ethics statement should only appear in the Methods section of your manuscript. If your ethics statement is written in any section besides the Methods, please delete it from any other section. It's only mentioned in the method section under the title of “Ethical approval”, page 6.

8 Please review your reference list to ensure that it is complete and correct. References had been checked.

List of Changes based on Manuscript Track Change (Reviewer 1):

No Reviewers’ comments Changes

1 Strengthen the rationale for a qualitative approach by highlighting gaps in previous quantitative or survey-based studies in the introduction part. In response, we have revised the introduction to strengthen the rationale for adopting a qualitative approach. Specifically, we have emphasized the limitations of previous quantitative and survey-based studies on pharmacovigilance systems, which often focus on numerical data such as reporting rates or awareness levels. These studies, while informative, do not sufficiently explore the contextual, organizational, and experiential factors that influence pharmacovigilance practices in healthcare settings. These revisions have been incorporated in the introduction section; page 5.

2 No details on the demographic distribution of public vs. private hospitals. Thank you for this observation. We have now added details clarifying the demographic distribution of the participating hospitals, including the representation of public and private institutions. Specifically, the study sample included [2] public hospitals and [8] private hospitals, selected to reflect the diversity of Dubai's healthcare system. This information has been incorporated into the Methods section under the description of study settings (page 6, paragraph 2), to provide clearer context for the institutional backgrounds of participants.

3 The discussion reiterates many points from the results section without sufficient theoretical grounding or linkage to wider literature. We have revised and rewritten the Discussion section to provide stronger theoretical grounding. Additional literature has been incorporated to better interpret the results and emphasize their contribution to the existing body of knowledge. Key points are now supported by citations to enhance the academic rigor of the discussion.

4 Explicitly mention limitations inherent to the design and context, such as the exclusion of non-hospital settings or regulators. Thank you for the comment. These limitations are mentioned under title of Limitations, page 30: including all these required points.

5 Ethics approval is mentioned, but there is no mention of how confidentiality was maintained during audio recording and transcription. We have updated the Methods section to clarify the steps taken to ensure participant confidentiality throughout the data collection and transcription process. Under section of: Ethical Approval, page 6.

List of Changes based on Manuscript Track Change (Reviewer 2):

No Reviewers’ comments Changes

1 The study uses convenience and purposive sampling limited to only 10 participants, mostly male, and primarily senior positions (chief pharmacists and managers). As noted, the study utilized purposive and convenience sampling, focusing on hospital pharmacists in leadership roles, such as chief pharmacists and pharmacy managers. This sampling strategy was chosen to target individuals with direct responsibility for or oversight of pharmacovigilance activities, ensuring that participants could provide informed insights into institutional processes, challenges, and practices. The predominance of male participants reflects, to some extent, the current gender distribution in senior pharmacy positions within the hospital settings studied.

2 The study focuses solely on hospitals in Dubai, limiting applicability to other Emirates or healthcare sectors in the UAE. This focus was intentional, as Dubai has a unique and diverse healthcare landscape that includes both public and private hospitals with relatively well-established pharmacovigilance frameworks. However, we agree that this geographical limitation may restrict the broader applicability of the findings to other Emirates or to non-hospital healthcare sectors such as primary care clinics, community pharmacies, or regulatory institutions.

It is mentioned in the Limitations section (page 30)

3 Underdeveloped Thematic Framework: Although seven themes are outlined, some subthemes are repetitive or overlapping, and coding transparency is minimal. - Subtheme 1 in Theme 3 has been merged with Subtheme 1 in Theme 1 under the new title: “Presence and familiarity of PV center or system in hospital”.

- Subtheme 4 in Theme 3 has been merged with Subtheme 2 in Theme 2 under the title of: “Type of ADR to be reported”.

- The rest of the subthemes fit in their themes as far as the author concerned. The above 2 subthemes had been merged based on the reviewer recommendations.

4 Language and Grammar Issues: There are numerous grammatical errors, awkward phrasing, and redundancies throughout the manuscript. Conduct professional language editing to enhance readability and academic tone. we have thoroughly revised the text to address grammatical errors, eliminate awkward phrasing, and remove redundancies. The entire manuscript has undergone professional language editing to improve readability, coherence, and academic tone.

5 Minimal Integration of Patient and Public Role: Although patient reporting is briefly mentioned, the manuscript lacks substantive exploration of patient or caregiver involvement in pharmacovigilance. We acknowledge the importance of patient and caregiver involvement in pharmacovigilance. However, this study specifically focused on exploring the role of hospital pharmacists in ADR reporting within institutional settings. As such, the perspectives and roles of patients or caregivers were outside the scope of the research objectives.

6 Ethics and Consent Process Description: While ethical approvals are mentioned, the informed consent process and interview setting details (e.g., potential coercion, power dynamics) are not fully elaborated. We have updated the Methods section to clarify the steps taken to ensure participant confidentiality throughout the data collection and transcription process. Under section of: Ethical Approval, page 6.

---

## [Editor Report · Decision Letter 1]

25 Aug 2025

A Qualitative Study to investigate pharmacovigilance systems in Dubai hospitals

PONE-D-25-10627R1

Dear Dr.Alomar,

We’re pleased to inform you that your manuscript has been judged scientifically suitable for publication and will be formally accepted for publication once it meets all outstanding technical requirements.

Kind regards,

Yaser Mohammed Al-Worafi

Academic Editor

PLOS ONE
---

## [Editor Report · Acceptance letter]

PONE-D-25-10627R1

PLOS ONE

Dear Dr. Alomar,

I'm pleased to inform you that your manuscript has been deemed suitable for publication in PLOS ONE. Congratulations! Your manuscript is now being handed over to our production team.

Kind regards,

on behalf of

Professor Yaser Mohammed Al-Worafi

Academic Editor

PLOS ONE